# Comparison of the 2013 and 2019 Nationwide Surveys on the Management of Chronic Kidney Disease by General Practitioners in Japan

**DOI:** 10.3390/jcm11164779

**Published:** 2022-08-16

**Authors:** Satoru Tatematsu, Kazuo Kobayashi, Yasunori Utsunomiya, Tsuguru Hatta, Taisuke Isozaki, Masanobu Miyazaki, Yosuke Nakayama, Takuo Kusumoto, Nobuo Hatori, Haruhisa Otani

**Affiliations:** 1Committee of Kidney and Electrolyte Disease, Japan Physicians Association, Tokyo 101-0062, Japan; 2Division of Nephrology, Department of Internal Medicine, Edogawa Medicare Hospital, Tokyo 133-0052, Japan; 3Department of Medical Science and Cardiorenal Medicine, Yokohama City University Graduate School of Medicine, Yokohama 236-0004, Japan; 4Kobayashi Hospital, Odawara 250-0011, Japan

**Keywords:** chronic kidney disease, general practitioner, questionnaire, survey, guidelines

## Abstract

In 2019, the Japan Physicians Association conducted a second nationwide survey on the management of chronic kidney disease (CKD) among the Japanese general practitioners (GPs). We aimed to clarify the changes in the state of CKD medical care by GPs since the 2013 survey. The 2013 and 2019 surveys included 2214 and 601 GPs, respectively, who voluntarily participated. The two surveys were compared, using propensity score matching to balance the background of the responded GPs. For the medical care of CKD, the frequency of urine or blood examination, use of estimated glomerular filtration rate (eGFR) value for CKD management, and continuous use of renin-angiotensin system inhibitors for their reno-protective effects were significantly higher in 2019 than in 2013 (all: *p* < 0.001). The medical cooperation in CKD management, the utilization of the clinical path for CKD management and the measurement of the eGFR during the medical health checkup were significantly increased in 2019, compared to those in 2013. More GPs felt dissatisfied with the components of CKD treatment by nephrologists (*p* < 0.001). The two surveys confirmed improvements in the level of medical care for CKD and a strengthening in cooperation. However, the dissatisfaction with the consultation with nephrologists did not necessarily improve.

## 1. Introduction

More than 850 million people globally [1] and more than 13.3 million people in Japan have chronic kidney disease (CKD) [2]. CKD has a high risk for end-stage kidney disease that is a leading cause of mortality. Liyanage et al. [3] estimated that the number of patients who will have received kidney replacement therapy will increase from 2.6 million in 2010 to 5.4 million in 2030. Furthermore, they also estimated that at least 2.284 million patients may have died prematurely in 2010 because kidney replacement therapy was not available. CKD is a common disease, and many patients with CKD consult general practitioners (GPs). Several campaigns for the promotion of CKD were conducted, and CKD guidebooks or guidelines [2,4] were published to improve the level of CKD management by GPs. To clarify the actual situation of medical care for CKD by GPs, we conducted the first nationwide survey in 2013. Our first survey clarified that GPs’ subspecialty and training history in nephrology substantially influenced CKD management and medical cooperation in Japan [5]. Two major events occurred in 2018 in Japan. First, the Japanese Ministry of Health, Labor and Welfare (Tokyo, Japan) published a report with the aim of further promoting measures against CKD (https://www.mhlw.go.jp/stf/shingi2/0000172968_00002.html, accessed on 1 January 2020). In the Ministry’s Report, the key performance indicators were set as government efforts for CKD measures, the reinforcement of cooperation between nephrologists and GPs, and a decrease of more than 10% in the number of patients who began kidney replacement therapy after 10 years. Second, in 2018, the Japanese Society of Nephrology (Tokyo, Japan) established the Japan Kidney Association, a nonprofit organization, to enlighten, disseminate, overcome diseases, and contribute to society. Based on the domestic activities related to CKD in Japan, we conducted a second nationwide survey in 2019. The aim of this study was to clarify the differences in the results of the nationwide surveys between 2013 and 2019.

## 2. Materials and Methods

### 2.1. Participants, Survey, and Procedures

The Japan Physicians Association, which is a nationwide organization with 15,000 GPs in hospitals or clinics in Japan, conducted this survey. We mailed the survey to all of the members of the Japan Physicians Association in August 2019. Among all of the members, 601 GPs (approximately 4.0%) who voluntarily answered between August and November 2019 participated in this survey. The present study involved the same 21 questions that were asked in 2013 and in 2019 for comparison. The survey consisted of two parts: the first part consisted of 10 questions regarding the diagnosis and management of CKD in clinical practice by the GPs and the second part consisted of 11 questions regarding the cooperation between GPs and nephrologists in their region, including the political measures for CKD management by the local government. This survey was conducted in accordance with the Declaration of Helsinki.

### 2.2. Propensity Score Matching

In the comparisons between the two groups in observational studies, confounding factors influenced the outcome, which can lead to serious bias. Propensity score (PS) matching is a statistical technique utilized to diminish bias due to the covariates in observational studies. We also collected the characteristics of the respondent GPs, including the age distribution, workplace, the population of a medical area, specialty, and nephrology training history in the 2013 and 2019 surveys. In this study, PS matching was utilized to match the respondent GPs with baseline characteristics. The PS for the GPs who were participants in the 2019 survey was calculated by using logistic regression analysis with the following characteristic variables: age distribution, workplace, the population of a medical area, specialty, and history of training in nephrology. The algorithm used in the PS matching in this study was a 1:1 nearest neighbor match with a 0.017 caliper value, which was equivalent to 0.2 times of the standard deviation of the PS [6]. There was no replacement when the matched model was built in this study.

### 2.3. Statistical Analysis

The characteristics of the GPs and the answers to the 21 questions were reformed to discrete variables. The Chi-squared test was used for comparisons between the two groups. Statistical significance was set at *p* < 0.05. All of the analyses were conducted using IBM SPSS Statistics software (version 25.0; IBM Inc., Armonk, NY, USA).

## 3. Results

### 3.1. Characteristics of the Participants

The characteristics of the respondent GPs of the surveys in 2013 and 2019 for the unmatched cohort and matched cohort models are shown in Table 1. In the unmatched model, the distribution of all of the characteristics of the respondent GPs, age distribution, workplace, the population of a medical area, specialty, and nephrology training history showed significant differences between the surveys in 2013 and 2019 with *p*-values of <0.001, <0.001, <0.001, <0.001, 0.02, and <0.001, respectively.

For the age distribution, the frequency of the older GPs was higher in the 2019 survey, and the respondent GPs > 50 years old constituted 81.8% (*n* = 1880) in the 2013 survey and 88.4% (*n* = 531) in the 2019 survey (*p* < 0.001). For the population of a medical area, the frequency of the GPs practicing where the population of a medical area was large was higher in the 2019 survey. The respondent GPs practicing in an area with >100 × 10^3^ individuals constituted 59.5% (*n* = 1381) in the 2013 survey and 69.4% (*n* = 417) in the 2019 survey (*p* < 0.001). The characteristics of the workplace, specialty, and nephrology training history were statistically different in the unmatched model (*p* < 0.001, respectively), and the ratio of the subspecialty of diabetology/endocrinology was significantly higher in the 2019 survey (*p* < 0.001). However, the distributions were similar, and whether such small differences had some impact on the results was unclear.

In the matched model, there were no significant differences between the two groups. The absolute standardized difference was utilized to evaluate the balance of the two groups on the matched model, and the values of <1.96 × √(2/*n*) for the measured covariates suggested an appropriate balance between the groups [7]. All of the standardized differences between the GPs’ characteristics were below 0.09, except for the subspecialty of diabetology/endocrinology, neurology, and collagen disease/rheumatology, which was under the estimated borderline in this matched model (*n* = 574 in each group, 1.96 × √(2/*n*) = 0.12). Appendix A shows well-balanced histograms of the characteristics of the two groups in the matched model.

Although a small difference was observed in the subspecialties of the respondent GPs, the age distributions and the proportion of GPs who work in clinics on the matched model were close to those of all of the members of the Japan Physicians Association. This survey may be typical of the GP population in Japan. However, the very low rate of the respondent GPs in the 2019 survey may induce the selection bias, and this is the major concern of this survey.

### 3.2. Comparison between the Surveys in 2013 and 2019 on the PS-Matched Model with 574 GPs in Each Group

A matched model with 574 GPs was analyzed to compare the results of the surveys in 2013 and 2019. The results of the questions for which the respondent GPs chose one answer limb are shown in Table 2 and Table 3, and the results of the questions for which respondent GPs could choose multiple answer limbs are shown in Figure 1. Table 2 and Figure 1 (question 9) shows the results of the questions on the diagnosis and management of CKD in clinical practice by GPs. The respondent GPs in 2019 knew less about the revision of the CKD guidelines (*p* < 0.02). The frequency of GPs who did not have or did not use them was larger in the 2019 survey than in the 2013 survey (69.0% (*n* = 396) and 61.5% (*n* = 353), respectively, *p* < 0.001).

A significant increase in the GPs who conducted urine and blood examinations, which are basic medical examinations for CKD management, was frequently observed in the 2019 survey, compared to the 2013 survey, based on the results of questions three and four (*p* < 0.001). More of the GPs used the value of the eGFR for CKD guidance in the 2019 survey than in the 2013 survey (*p* < 0.001); however, the frequency of checking the quantification of proteinuria was lower in the 2019 survey than in the 2013 survey (*p* = 0.01).

The BP measurement in the office and the use of an erythropoietin-stimulating agent for patients with CKD showed no significant differences between the two surveys (*p* = 0.10 and *p* = 0.07, respectively). Compared to the GPs in the 2013 survey, more of the GPs in the 2019 survey continuously used renin-angiotensin system (RAS) inhibitors for their reno-protective effects (*p* < 0.01). Furthermore, the number of GPs who discontinued RAS inhibitors when the serum creatinine exceeded 2 mg/dL did not change, and the frequency was relatively small (25.6% in the 2013 survey and 21.4% in the 2019 survey in the PS-matched model). Meanwhile, the number of GPs who discontinued the RAS inhibitors when the serum potassium level rose over the normal range was significantly increased in the 2019 survey, compared to that in the 2013 survey (*p* = 0.001 and *p* < 0.001, respectively).

Table 3 and Figure 1 show the results of the questions for the cooperation between GPs and nephrologists in their region, including the political measures for CKD management by the local government. No significant reinforcement existed regarding improvement in the regional cooperation for CKD management, including the personal friendship between GPs and nephrologists, from 2013 to 2019, based on the results of questions 11 and 12 (*p* = 0.38 and *p* = 0.08, respectively). The proportion of the GPs who answered that the clinical pathway for the management of CKD patients was working in their region increased from 3.8% in the 2013 survey to 16.0% in the 2019 survey. However, most of the (81.7%) GPs neither established nor used them in their medical area in 2019. The frequency of consultations with nephrologists for any reason, except for the control of BP or blood glucose, increased in the 2019 survey, compared to that in the 2013 survey. The frequent answer regarding the stage of eGFR at which GPs consulted with nephrologists was stage G3b (45 > eGFR ≥ 30 mL/min/1.73 m^2^) in both of the surveys in 2013 and 2019. However, fewer GPs in the 2019 survey consulted with nephrologists for patients with stage G2 (90 > eGFR ≥ 60 mL/min/1.73 m^2^) or G3a (60 > eGFR ≥ 45 mL/min/1.73 m^2^), compared to that in the 2013 survey (*p* = 0.01 and *p* < 0.001, respectively).

Regarding requests from GPs to nephrologists, confirmation of CKD treatment and improvement in eGFR, correction of electrolyte imbalance, counseling by nephrologists of CKD patients, and counseling by nutritionists of CKD patients were significantly increased in the 2019 survey than those in the 2013 survey (*p* = 0.03, *p* < 0.001, *p* = 0.003, and *p* = 0.01, respectively). However, the GPs who requested renal biopsy or digital examination had significantly decreased in the 2019 survey than those in the 2013 survey (*p* < 0.001). The ratio of the GPs who felt satisfied with the nephrologist’s consultation did not significantly change (*p* = 0.32), although more GPs felt dissatisfied with the counseling or explanation to CKD patients or the components of CKD treatment by nephrologists (*p* = 0.02 and *p* < 0.001, respectively) in the 2019 survey than in the 2013 survey.

## 4. Discussion

The aim of this study was to clarify the change in the state of CKD medical care by GPs, based on information derived from the surveys conducted in 2013 and 2019. The rates of awareness, possession, and use of the revised guidelines were lower in 2019 than those in 2013. Several of the studies demonstrated that GP awareness and management of CKD are insufficient, and that GPs generally underrecognize and undertreat CKD [8,9,10,11,12,13]. In practice, multiple barriers exist to utilizing the CKD guidelines [14]. Common barriers are a lack of time, fear of communicating a diagnosis of CKD to a patient, and dissatisfaction with the CKD guidelines. It is important to analyze the various factors that hinder the use of the clinical practice guidelines and the dissemination methods for clinicians who need medical practice guidelines to use them appropriately. The Japanese Society of Nephrology established the Japan Kidney Association, a nonprofit organization, in 2018 to enlighten, disseminate, overcome diseases, and contribute to society. As one of its main projects, the CKD Countermeasures Subcommittee, which is responsible for CKD countermeasures in line with the actual conditions of the region, appointed a person in charge in each prefecture and disseminated JSN 2018 to GPs in a manner that suited the actual conditions of each region, and is building a medical-care cooperation system.

The quantification of urine protein is indispensable for the management of CKD. In the United States, the rate of urinary protein quantification is 30% in patients with stage 3 or 4 CKD [15]. In the Netherlands, the rate of urinary albumin quantification in stage 3 CKD patients is 40% [16]. In Canada, 42% of patients with an initial albumin/creatinine ratio (ACR) above 3 mg/mmol had an ACR test again during the next six months, and 16% of patients with an initial eGFR of <60 mL/min/1.73 m^2^ had an ACR test over the next six months [17,18]. In Australia, the albuminuria testing rate for stage 3 CKD patients was less than 50% [19]. In this survey, the rate of quantitative urinary protein testing was 76% in 2013 and 65% in 2019. The implementation rate of urinary protein quantification, although it cannot be simply compared, is higher in Japan than in other countries, based on the results. However, because urinary protein quantification is inevitable in CKD clinical practice, the implementation rate of the test should be close to 100%. In other countries, improvements in CKD practice have been reported over time. In China, 1999–2000 and 2004–2005 nationwide surveys showed improvements in hypertension awareness, treatment, and control among CKD patients [20]. In the UK, individuals were assessed both in 2010 and at the date of their first classification of CKD in the General Practice Research Database [21]. When the patients were stratified by date of diagnosis, the proportion of patients with stage 3–5 CKD and cardiovascular-related comorbidities decreased with time, and the use of lipid-modifying agents and antihypertensives in patients with dyslipidemia and hypertension increased with time. Those results suggest that the introduction of CKD into the Quality and Outcomes Framework, which is a system for the performance management and payment of GPs, has increased the awareness of CKD among physicians in the UK, allowing for earlier intervention and a better control of CKD progression [21]. It is presumed that CKD management has become widely recognized over time not only in Japan but also in other countries, due to the efforts and enlightenment activities of physicians.

In the PS-matched model in this study, the number of GPs who continued to administer RAS inhibitors to CKD patients for as long as possible in anticipation of renoprotection significantly increased. Generally, the change in the administration of RAS inhibitors is considered to be favorable. In one report [22], the eGFR increased from 16.4 ± 1.0 mL/min/1.73 m^2^ to 26.6 ± 2.2 mL/min/1.73 m^2^ at one year after an angiotensin-converting enzyme (ACE) inhibitor or angiotensin II receptor blocker (ARB) was replaced with another antihypertensive agent, such as a calcium channel blocker in stage G4 or -G5 CKD patients, suggesting that the discontinuation of RAS inhibitors can delay the onset of kidney replacement therapy in the advanced CKD patients. Other studies show that a 10–30% decrease in eGFR or a 10–30% increase in serum creatinine within two months after starting an ACE inhibitor or ARB leads to the long-term deterioration of renal function [23,24]. Thus, GPs are required to discontinue RAS inhibitors in CKD patients at a proper time. In future, we need to investigate in more detail whether the GPs discontinue the RAS inhibitors at the right time.

The judgment by the GPs for referral to nephrologists improved in 2019, compared to that in 2013. Regarding medical collaboration, in the conventional referral criteria from GPs to nephrologists, CKD stage G3A1 was divided into three categories (i.e., 30–39 mL/min/1.73 m^2^, 40–49 mL/min/1.73 m^2^, and 50–59 mL/min/1.73 m^2^), and patients were referred based on age {nephrology, 2012 #439} [2], which was complicated for users. Therefore, by referring to the data on medical checkups aimed at preventing lifestyle-related diseases [25], the committee members of the Japanese Society of Nephrology reviewed the standard eGFR and age to simplify the referral criteria, so that appropriate medical collaboration could be started in a timely manner. A presumption is that the simplification of the CKD guidelines improved the referrals to nephrologists. The results of our other analysis of 2019 survey showed that the familiarity with nephrologists is related to the appropriate cooperation for CKD management (data not shown). We suspect that promoting face-to-face collaboration with nephrologists and GPs and the further utilization of the CKD guidelines will lead to appropriate CKD management in the future.

This study has some limitations. First, we could not eliminate the possibility of selection bias, because the response rates were very low (4% in 2019 survey), and the respondents were different between the 2013 and 2019 groups. In the result of the subspecialty, the relatively large values of the standardized differences were observed in the subspecialty of diabetology/endocrinology, neurology, and collagen disease/rheumatology. In particular, the larger rate of the respondent GPs with the subspeciality of diabetology/endocrinology may influence the results. Second, the generalizability of these findings is limited because the sample size was small, especially after PS matching. The PS matching method was useful for improving the imbalance of the characteristics of the respondent GPs in this study. However, only matched GPs were included in the comparison analysis, and the results did not necessarily apply to the GPs who were excluded after PS matching.

Regarding the difference of the numbers of the respondent GPs in the 2013 and 2019 survey, the subjects of the survey were not necessarily the same. In the 2019 survey, we mailed the survey to all of the members (15,000) of the Japan Physicians Association. In the 2013 survey, in addition to all of our members, we also distributed the survey to 12,400 GPs who were not our members, for example, the members of the Local Physicians Association. Therefore, the respondent rate in the 2013 survey was 8.5%, and 1364 of the 2320 (59%) respondents were our members. Furthermore, the 2013 survey was the first nationwide survey, and a big promotion or announcement of the survey was performed. These can influence the difference in the numbers of the respondent GPs in the 2013 and 2019 survey.

## 5. Conclusions

Through the two surveys, improvements in the level of medical care for CKD and the strengthening of cooperation were confirmed. However, the dissatisfaction with the consultations with nephrologists did not necessarily improve, and the decrease in the possession or utilization of the CKD guidelines is a future issue for investigation. We will evaluate the effects of the attempts by the Japanese Society of Nephrology henceforth.

## 6. Compliance with Ethical Standards

### 6.1. Research Involving Human Participants and/or Animals

Approval by an ethics committee was not required, because this survey was conducted among medical doctors, and did not directly involve any patients, and samples collected from the human body were not used.

### 6.2. Informed Consent

The survey was administered among the members of the Japan Physicians Association. Explanations about the implementation of the survey, the presentation of the results, and the writing of the manuscript were written on the survey sheet. The responses to the survey provided informed consent.

## Figures and Tables

**Figure 1 jcm-11-04779-f001:**
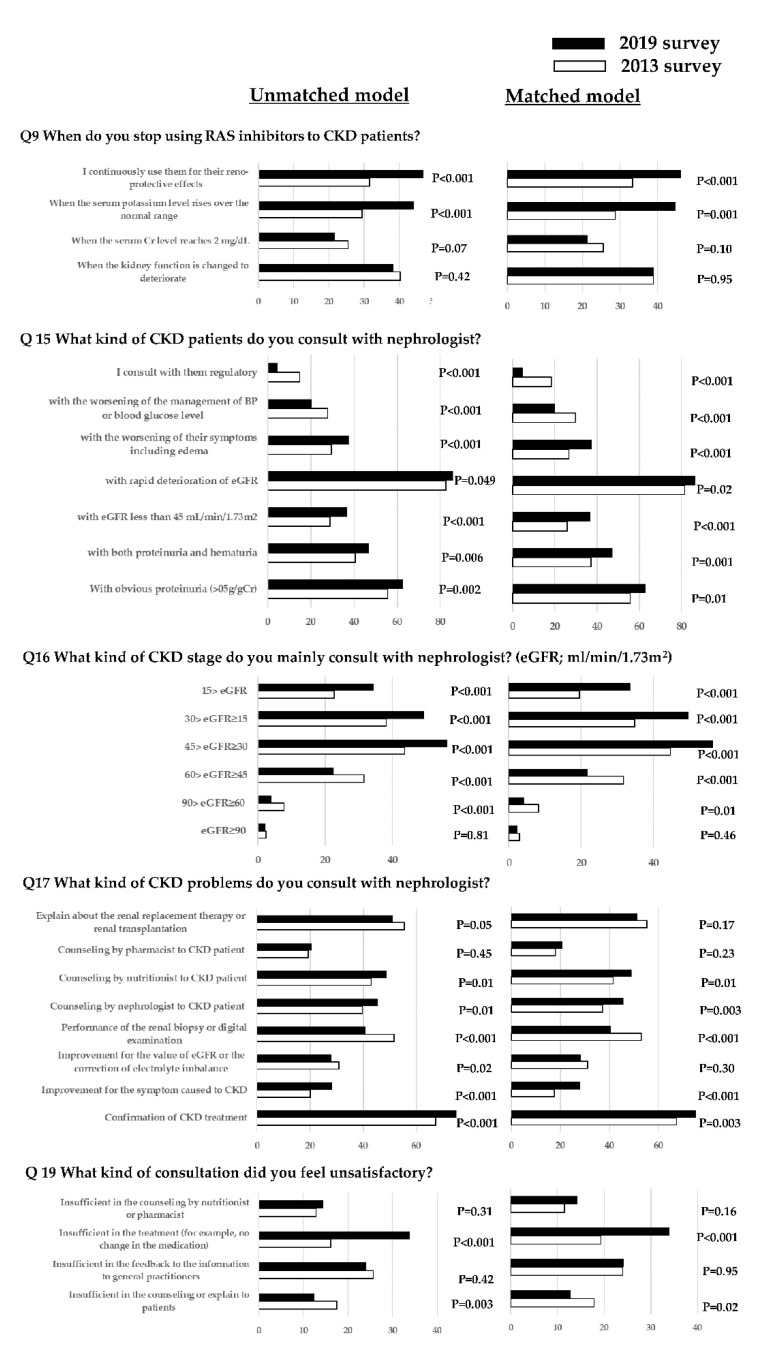
Questionnaire results of the questions for which respondent GPs could choose multiple answer limbs on the survey in 2013 and 2019 for the unmatched and matched cohort models.

**Table 1 jcm-11-04779-t001:** Characteristics of the respondent GPs of the survey in 2013 and 2019 for unmatched and matched cohort models.

			Unmatched Model			Matched Model			
		Total (*n* = 2921)	Survey in 2013 (*n* = 2320)	Survey in 2019 (*n* = 601)	*p*-Value	Survey in 2013 (*n* = 574)	Survey in 2019 (*n* = 574)	*p*-Value	Standardized Difference
Age distribution	20s	6 (0.2%)	3 (0.1%)	3 (0.5%)	<0.001	3 (0.5%)	1 (0.2%)	0.95	0.06
	30s	92 (3.1%)	78 (3.4%)	14 (2.3%)		12 (2.1%)	14 (2.4%)		0.02
	40s	395 (13.5%)	345 (14.9%)	50 (8.3%)		53 (9.2%)	50 (8.7%)		0.02
	50s	1025 (13.5%)	854 (14.9%)	171 (28.5%)		170 (29.6%)	169 (29.4%)		0.004
	60s	839 (28.7%)	628 (27.1%)	211 (35.1%)		203 (35.4%)	204 (35.5%)		0.004
	over 70	547 (18.7%)	398 (17.2%)	149 (24.8%)		132 (23.0%)	134 (23.3%)		0.008
	No answer	17 (0.6%)	14 (0.6%)	3 (0.5%)		1 (0.2%)	2 (0.3%)		0.03
Workplace	Clinic	2513 (86.0%)	1996 (86.0%)	517 (86.0%)	<0.001	527 (91.8%)	514 (89.5%)	0.40	0.08
	Hospital	278 (9.5%)	205 (8.8%)	73 (12.1%)		41 (7.1%)	51 (8.9%)		006
	No answer	130 (4.5%)	119 (5.1%)	11 (1.8%)		6 (1.0%)	9 (1.6%)		0.05
Population of a medical area	~10 × 10^3^	197 (6.7%)	162 (7.0%)	35 (5.8%)	<0.001	32 (5.6%)	35 (6.1%)	0.96	0.02
	10 × 10^3^–50 × 10^3^	444 (15.2%)	359 (15.5%)	85 (14.1%)		80 (13.9%)	83 (14.5%)		0.01
	50 × 10^3^–100 × 10^3^	395 (13.5%)	335 (14.4%)	60 (10.0%)		61 (10.6%)	60 (10.5%)		0.006
	100 × 10^3^–500 × 10^3^	1115 (38.2%)	832 (35.9%)	283 (47.1%)		272 (47.4%)	263 (45.8%)		0.03
	500 × 10^3^-	683 (23.4%)	549 (23.7%)	134 (22.3%)		124 (21.6%)	130 (22.6%)		0.03
	No answer	87 (3.0%)	83 (%3.6)	4 (0.7%)		5 (0.9%)	3 (0.5%)		0.04
Specialty	Nephrologist	371 (12.7%)	298 (12.8%)	73 (12.1%)	0.02	56 (9.8%)	63 (11.0%)	0.80	0.04
	Non-nephrologist	2546 (87.2%)	2021 (87.1%)	525 (87.4%)		517 (90.1%)	510 (88.9%)		0.04
	No answer	4 (0.1%)	1 (0.0%)	3 (0.5%)		1 (0.2%)	1 (0.2%)		0.0
Subspecialty (Multiple answers are available)	General internal medicine	2551 (87.3%)	2030 (87.5%)	521 (86.7%)	0.59	513 (89.4%)	502 (87.5%)	0.31	0.09
	Nephrology	371 (12.7%)	297 (12.8%)	74 (12.3%)	0.75	56 (9.8%)	64 (11.1%)	0.44	0.07
	Cardiology	650 (22.3%)	527 (22.7%)	123 (20.5%)	0.24	131 (22.8%)	117 (20.4%)	0.32	0.07
	Diabetology/Endocrinology	397 (13.6%)	291 (12.5%)	106 (17.6%)	0.001	74 (12.9%)	101 (17.6%)	0.03	0.18
	Gastroenterology	705 (24.1%)	560 (24.1%)	145 (24.1%)	0.995	151 (26.3%)	142 (24.7%)	0.54	0.04
	Pulmonology	253 (8.7%)	203 (8.8%)	50 (8.3%)	0.74	48 (8.4%)	46 (8.0%)	0.83	0.02
	Neurology	62 (2.1%)	53 (2.3%)	9 (1.5%)	0.23	14 (2.4%)	9 (1.6%)	0.29	0.22
	Neurosurgery	30 (1.0%)	22 (0.9%)	8 (1.3%)	0.41	5 (0.9%)	6 (1.0%)	0.76	0.09
	Hematology	35 (1.2%)	26 (1.1%)	9 (1.5%)	0.45	8 (1.4%)	9 (1.6%)	0.81	0.06
	Collagen disease /Rheumatology	76 (2.6%)	58 (2.5%)	18 (3.0%)	0.50	11 (1.9%)	16 (2.8%)	0.33	0.19
	Allergology	100 (3.4%)	78 (3.4%)	22 (3.7%)	0.72	22 (3.8%)	22 (3.8%)	1.0	0.0
	Others (except for internal medicine)	276 (9.4%)	220 (9.5%)	56 (9.3%)	0.90	52 (9.1%)	55 (9.6%)	0.76	0.03
History of training of nephrology	Yes	1000 (34.2%)	795 (34.3%)	205 (34.1%)	<0.001	192 (33.4%)	189 (32.9%)	0.98	0.01
	no	1911 (65.4%)	1523 (65.6%)	388 (64.6%)		380 (66.2%)	383 (66.7%)		0.01
	No answer	10 (0.3%)	2 (0.1%)	8 (1.3%)		2 (0.3%)	2 (0.3%)		0.0

Values are *n*/total *n* (%). *p* values by chi-squared test. Abbreviations: GP: general practitioner.

**Table 2 jcm-11-04779-t002:** Questionnaire results of the respondent GPs of the survey in 2013 and 2019 for the unmatched and matched cohort models; questions for the diagnosis and management of CKD in clinical practice.

			Unmatched Model			Matched Model		
		Total (*n* = 2921)	Survey in 2013 (*n* = 2320)	Survey in 2019 (*n* = 601)	*p*-Value	Survey in 2013 (*n* = 574)	Survey in 2019 (*n* = 574)	*p*-Value
Q1	Did you know that CKD guidelines were revised last year?							
	No	324 (11.1%)	230 (9.9%)	94 (15.6%)	0.001	59 (10.3%)	92 (16.0%)	0.02
	I knew them	1519 (52.0%)	1203 (51.9%)	316 (52.6%)		306 (53.3%)	304 (53.0%)	
	I knew and recognized them	1068 (36.6%)	881 (38.0%)	187 (31.1%)		207 (36.1%)	175 (30.5%)	
	No answer	10 (0.3%)	6 (0.3%)	4 (0.7%)		2 (0.3%)	3 (0.5%)	
Q2	Do you have CKD guidelines?							
	No	521 (17.8%)	311 (13.4%)	210 (34.9%)	0.001	83 (14.5%)	202 (35.2%)	<0.001
	Yes, but I did not use them	1301 (44.5%)	1097 (47.3%)	204 (33.9%)		270 (47.0%)	194 (33.8%)	
	Yes, and I use frequently	1064 (36.4%)	893 (38.5%)	171 (28.5%)		217 (37.8%)	163 (28.4%)	
	No answer	35 (1.2%)	19 (0.8%)	16 (2.7%)		4 (0.7%)	15 (2.6%)	
Q3	Do you check the urine analysis of CKD patients during a regular visit?							
	I always check it	1185 (40.6%)	780 (33.6%)	405 (67.4%)	<0.001	191 (33.3%)	387 (67.4%)	<0.001
	I check it as needed	1522 (52.1%)	1398 (60.3%)	124 (20.6%)		349 (60.8%)	120 (20.9%)	
	I do not check it during a regulatory visit	197 (6.7%)	134 (5.8%)	63 (10.5%)		31 (5.4%)	59 (10.3%)	
	No answer	17 (0.6%)	8 (0.3%)	9 (1.5%)		8 (1.4%)	3 (0.5%)	
Q4	Do you check the quantification of proteinuria?							
	Yes	2145 (73.4%)	1753 (75.6%)	392 (65.2%)	<0.001	418 (72.8%)	372 (64.8%)	0.01
	No	750 (25.7%)	548 (23.6%)	202 (33.6%)		149 (26.0%)	195 (34.0%)	
	No answer	26 (0.9%)	19 (0.8%)	7 (1.2%)		7 (1.2%)	7 (1.2%)	
Q5	How often do you check blood examination of CKD patients?							
	Every 1–4 months	491 (16.8%)	289 (12.5%)	202 (33.6%)	<0.001	73 (12.7%)	192 (33.4%)	<0.001
	Once or twice a year	2346 (80.3%)	1962 (84.6%)	384 (63.9%)		485 (84.5%)	368 (64.1%)	
	I do not check blood examination	22 (0.8%)	20 (0.9%)	2 (0.3%)		7 (1.2%)	2 (0.3%)	
	No answer	62 (2.1%)	49 (2.1%)	2 (2.2%)		9 (1.6%)	12 (2.1%)	
Q6	Do you use the value of eGFR on the CKD guidance?							
	I often use it	1480 (50.7%)	1042 (44.9%)	438 (72.9%)	<0.001	263 (45.8%)	420 (73.2%)	<0.001
	I sometimes use it	1075 (36.8%)	930 (40.1%)	145 (24.1%)		226 (39.4%)	137 (23.9%)	
	I do not use it	344 (11.8%)	331 (14.3%)	13 (2.2%)		82 (14.3%)	12 (2.1%)	
	No answer	22 (0.8%)	17 (0.7%)	5 (0.8%)		3 (0.5%)	5 (0.9%)	
Q7	Do you check the value of cystatin C?							
	I often check it	93 (3.2%)	79 (3.4%)	14 (2.4%)	0.001	11 (1.9%)	11 (1.9%)	0.04
	I sometimes check it	791 (27.1%)	601 (25.9%)	190 (31.6%)		145 (25.3%)	179 (31.2%)	
	I do not check it	2022 (69.2%)	1632 (70.3%)	390 (64.9%)		416 (72.5%)	377 (65.7%)	
	No answer	15 (0.5%)	8 (0.3%)	7 (1.2%)		2 (0.3%)	7 (1.2%)	
Q8	Do you check the blood pressure of CKD patients when they visit you?							
	I check the blood pressure both at the office and at home in the early morning	1677 (57.4%)	1314 (56.6%)	363 (60.4%)	0.28	308 (53.7%)	349 (60.8%)	0.10
	I check the blood pressure only at the office	1144 (39.2%)	926 (39.9%)	218 (36.3%)		242 (42.2%)	207 (36.1%)	
	I do not check it	13 (0.4%)	9 (0.4%)	4 (0.7%)		5 (0.9%)	3 (0.5%)	
	No answer	87 (3.0%)	71 (3.1%)	16 (2.7%)		19 (3.3%)	15 (2.6%)	
Q10	Do you use an erythropoietin-stimulating agent for CKD patients?							
	Yes, I use it with my own judgement	1735 (59.4%)	1377 (59.4%)	358 (59.6%)	0.10	329 (57.3%)	338 (58.9%)	0.07
	Yes, I use it when the nephrologist recommends	576 (19.7%)	466 (20.1%)	110 (18.3%)		117 (20.4%)	107 (18.6%)	
	No	581 (19.9%)	459 (19.8%)	122 (20.3%)		126 (22.0%)	118 (20.6%)	
	No answer	29 (1.0%)	18 (0.8%)	11 (1.8%)		2 (0.3%)	11 (1.9%)	

Abbreviations; CKD: chronic kidney disease; eGFR: estimated glomerular filtration rate; GP: general practitioner; RAS: renin-angiotensin system.

**Table 3 jcm-11-04779-t003:** Questionnaire results of the respondent GPs of the survey in 2013 and 2019 for the unmatched and matched cohort models: questions for the cooperation between GPs and nephrologist in their region including the political measures for CKD management by the local government.

			Unmatched Model			Matched Model		
		Total (*n* = 2921)	Survey in 2013 (*n* = 2320)	Survey in 2019 (*n* = 601)	*p*-Value	Survey in 2013 (*n* = 574)	Survey in 2019 (*n* = 574)	*p*-Value
Q11	Is the regional corporation for CKD working in your region?							
	No	621 (21.3%)	485 (20.9%)	136 (22.6%)	0.22	128 (22.3%)	130 (22.6%)	0.38
	Yes, partially	1573 (53.9%)	1250 (53.9%)	323 (53.7%)		308 (53.7%)	304 (53.0%)	
	Yes, enough	700 (24.0%)	567 (24.4%)	133 (22.1%)		135 (23.5%)	131 (22.8%)	
	No answer	27 (0.9%)	18 (0.8%)	9 (1.5%)		3 (0.5%)	9 (1.6%)	
Q12	Do you have any nephrologist whom you can consult about your CKD patients?							
	No	180 (6.2%)	125 (5.4%)	55 (9.2%)	<0.001	33 (5.7%)	50 (8.7%)	0.08
	Yes, I have only one nephrologist for consultation	825 (28.2%)	651 (28.1%)	174 (29.0%)		170 (29.6%)	172 (30.0%)	
	Yes, I have many nephrologists for consultation	1802 (61.7%)	1439 (62.0%)	363 (60.4%)		353 (61.5%)	343 (59.8%)	
	No answer	114 (3.9%)	105 (4.5%)	9 (1.5%)		18 (3.1%)	9 (1.6%)	
Q13	How is the relationship between you and the nephrologist whom you can consult?							
	I did not know well about them	613 (21.0%)	531 (22.9%)	82 (13.6%)	<0.001	130 (22.6%)	77 (13.4%)	<0.001
	I know only their name	1197 (41.0%)	860 (37.1%)	337 (56.1%)		231 (40.2%)	324 (56.4%)	
	I know them well as friend	1022 (35.0%)	849 (36.6%)	173 (28.8%)		204 (35.5%)	164 (28.6%)	
	No answer	89 (3.0%)	80 (3.4%)	9 (1.5%)		9 (1.6%)	9 (1.6%)	
Q14	Does your region have the clinical path for the management of CKD patients?							
	No	1912 (65.5%)	1561 (67.3%)	351 (58.4%)	<0.001	384 (68.6%)	335 (58.4%)	<0.001
	Yes, but it doesn’t work	757 (25.9%)	614 (26.5%)	143 (23.8%)		154 (26.8%)	134 (23.3%)	
	Yes, it is working now	206 (7.1%)	112 (4.8%)	94 (15.6%)		22 (3.8%)	92 (16.0%)	
	No answer	46 (1.6%)	33 (1.4%)	13 (2.2%)		4 (0.7%)	13 (2.3%)	
Q18	Did you satisfy the response for CKD consultation by nephrologist?							
	No	187 (6.4%)	152 (6.6%)	35 (5.8%)	0.01	43 (7.5%)	34 (5.9%)	0.32
	Neither	756 (25.8%)	615 (26.5%)	141 (23.5%)		150 (26.1%)	134 (23.3%)	
	Yes	1839 (63.0%)	1430 (61.6%)	409 (68.1%)		361 (62.9%)	390 (67.9%)	
	No answer	139 (4.8%)	123 (5.3%)	16 (2.7%)		20 (3.5%)	16 (2.8%)	
Q20	Is the measurement of serum creatinine included in the annual health check in your region?							
	No	500 (17.1%)	475 (20.5%)	25 (4.2%)	<0.001	118 (20.6%)	25 (4.4%)	<0.001
	Yes	2316 (79.3%)	1776 (76.6%)	540 (89.9%)		442 (77.0%)	514 (89.5%)	
	No answer	105 (3.6%)	69 (3.0%)	36 (6.0%)		14 (2.4%)	35 (6.1%)	
Q21	Is the health guidance for CKD implanted in your region?							
	No	650 (22.3%)	551 (23.8%)	99 (16.5%)	<0.001	125 (21.8%)	95 (16.6%)	<0.001
	Yes	542 (18.6%)	339 (14.6%)	203 (33.8%)		78 (13.6%)	196 (34.1%)	
	No answer	1729 (59.2%)	1430 (61.6%)	299 (49.8%)		371 (54.6%)	283 (49.3%)	

Abbreviations: CKD: chronic kidney disease; eGFR: estimated glomerular filtration rate; GP: general practitioner; RAS: renin-angiotensin system.

## Data Availability

The data were saved in a repository of the Japan Physicians Association. The datasets utilized and analyzed during the present study are available from the corresponding author upon reasonable request.

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
