# Peer review of "Comparison of the 2013 and 2019 Nationwide Surveys on the Management of Chronic Kidney Disease by General Practitioners in Japan"

_jcm, 2022, doi:10.3390/jcm11164779_

Round 1
Reviewer 1 Report
The authors compared the 2013 and 2019 surveys of a nationwide questionnaire on the management of chronic kidney disease by general practitioners in Japan. To my surprise, only 601 GPs (approximately 0.4%) were involved in the survey in 2019, which was much less than in 2013 and might lead to sample bias. The detailed departments of GPs should be considered. The main results were not significant enough, so the interest of the readers may be affected. Some paired results could be shown as figures other than tables, which could be clearer.
Author Response
Thank you for your important comment.
First, we apologize that we mistook the percentage of the respondent rate of the questionnaires; 0.4% was incorrect, and 4.0% was correct. Even if the rate was 4%, this was very low respondent rate. This low respondent rate may induce the selection bias and this is the major concern of this survey, as you mentioned. We added the information of the subspecialties of the respondent GPs in Table 1. The relatively large values of the standardized differences were observed in the subspecialty of diabetology/endocrinology, neurology, and collagen disease/rheumatology. Especially, the larger rate of the respondent GPs with the subspeciality of diabetology/endocrinology may influence the results.
We remade the Table 1 that includes the data of the subspecialty of the respondent GPs and added these sentences in Discussion section to emphasize the study limitation as below
First, we could not eliminate the possibility of selection bias because the response rates were very low (4% in 2019 survey), and the respondents were different between the 2013 and 2019 groups. In the result of the subspecialty, the relatively large values of the standardized differences were observed in the subspecialty of diabetology/endocrinology, neurology, and collagen disease/rheumatology. Especially, the larger rate of the respondent GPs with the subspeciality of diabetology/endocrinology may influence the results.
Also, according to your recommendation, we made Figure 1 to show the results. The results of the questions that respondent GPs chose one answer limb are shown in Table 2 and Table 3, and the results of the questions that respondent GPs could choose multiple answer limbs are shown in Figure 1.
We added the Figure 1 and some sentences in the Method section as follows;
The results of the questions that respondent GPs chose one answer limb are shown in Table 2 and Table 3, and the results of the questions that respondent GPs could choose multiple answer limbs are shown in Figure 1.
Reviewer 2 Report
The manuscript is well structured and reflects the current state of the issue as well as future perspectives.
The study is really interesting, above all because the survey is a comparison with previous ones from 2013. All Nephrology Societies ask for collaboration from the Associations of Family Doctors. Prevention of CKD is really important to be able to reduce the incidence of dialysis and transplantation.
In reality the paper has biases, but the statistical analysis is really important and well done.
As the authors comment, "However, dissatisfaction with consultation with nephrologists did not necessarily improve, and the decrease in the possession or utilization of CKD guidelines is a future issue for investigation".
Author Response
Thank you for your useful comment.
As your comments, the cooperation with the nephrologists and GPs are important for CKD management in clinical practice. The results of our other national questionnaires survey showed that the familiarity with nephrologists is related to the appropriate cooperation for CKD management (data were not shown). We suspect that promoting face-to-face collaboration with nephrologists and GPs and further utilization of CKD guidelines will lead to appropriate CKD management in the future. We added these sentences in Discussion section as below;
The results of other analysis of 2019 survey showed that the familiarity with nephrologists is related to the appropriate cooperation for CKD management (data were not shown). We suspect that promoting face-to-face collaboration with nephrologists and GPs and further utilization of CKD guidelines will lead to appropriate CKD management in the future.
Reviewer 3 Report
Dear authors,
your paper „Comparison of 2013 and 2019 surveys of a nationwide questionnaire on the management of chronic kidney disease by general practitioners in Japan” is written in a very informative and profound analysis on the quality of supply for CKD pts. in Japan. Relevant aspects and details have been addressed by the authors and the paper is written in a very comprehensible style. However, the paper has a strong focus on Japanese circumstances. Thus, in addition to the given discussion on quantification of urine protein (please see page 8, 1st chapter) it would be informative and interesting to compare detected results of the given survey, e.g. with the usage of RAS inhibitors or awareness of the diagnosis and treatment of CKD in other countries.
Minor aspects:
# page 3, line 104: it should be …>100x103 … instead of > 100x103
Author Response
Thank you for noting this concern. As you mentioned, the comparison between countries will be interesting and important for our discussion. However, it is difficult to compare our results with other countries because the similar questionnaire survey in the foreign countries cannot be found. There are some reports that related to the CKD management in clinical practice, we added some discussions as follows;
In other countries, improvements in CKD practice have been reported over time. In China, 1999-2000 and 2004-2005 nationwide surveys showed improvements in hypertension awareness, treatment and control among CKD patients [20]. In the UK, individuals were assessed both in 2010 and at the date of their first classification of CKD in the General Practice Research Database [21]. When patients were stratified by date of diagnosis, the proportion of patients with stage 3-5 CKD and cardiovascular-related comorbidities decreased with time, and the use of lipid-modifying agents and antihypertensives in patients with dyslipidemia and hypertension increased with time. Those results suggest that the introduction of CKD into the Quality and Outcomes Framework, which is a system for the performance management and payment of GPs, has increased awareness of CKD among physicians in the UK, allowing for earlier intervention and better control of CKD progression [21]. It is presumed that CKD management has become widely recognized over time not only in Japan but also in other countries due to the efforts and enlightenment activities of physicians.